# Medical challenges and unmet needs of individuals with 22q11.2 deletion syndrome as perceived by caregivers: A thematic analysis and natural language processing-based thematic extraction

Yutaka Sawai[1,2], Miho Tanaka[1], Shimon Tonsho[1], Akito Uno[1], Yusuke Takahashi[1], Akiko Kanehara[1], Yousuke Kumakura[1], Sho Yaghishita[3], Kiyoto Kasai[1,4]*

1 Department of Neuropsychiatry, Graduate School of Medicine, The University of Tokyo, Bunkyo-ku, Tokyo, Japan, 2 Unit for Mental Health Promotion, Research Center for Social Science & Medicine, Tokyo Metropolitan Institute of Medical Science, Tokyo, Japan, 3 Department of Structural Physiology, Graduate School of Medicine, The University of Tokyo, Bunkyo-ku, Tokyo, Japan, 4 The International Research Center for Neurointelligence (WPI-IRCN) at the University of Tokyo Institutes for Advanced Study (UTIAS), Bunkyo-ku, Tokyo, Japan

* kasaik-tky@g.ecc.u-tokyo.ac.jp

## Abstract

22q11.2 deletion syndrome (22q11DS) is associated with a variety of complications, including mental illness, intellectual disability, and physical disorders. Due to the overlap of these conditions, there is often a mismatch in existing healthcare frameworks, leading to unmet support needs. However, little is known about how patients and their caregivers perceive these issues. This study aims to automate thematic analysis (TA) and topic classification via natural language processing (NLP) techniques to extract medical needs from qualitative data provided by patients' caregivers. A web-based survey was conducted targeting caregivers of individuals with 22q11DS in Japan. 125 caregivers participated in the study and their responses were analyzed to identify medical challenges and unmet needs. TA and NLP-based thematic extraction was implemented on free-text responses related to medical concerns. To ensure privacy, the analysis was conducted offline using the open-source large language model (Cohere Command R Plus). Ethical considerations were addressed following the Declaration of Helsinki, with approval from the Ethics Committee of the University of Tokyo Graduate School of Medicine and Faculty of Medicine. NLP unveiled medical challenges and unmet needs of individuals with 22q11DS: a child-centered approach, comprehensive support across medical and welfare services, and support for caregivers through social and community networks. A comparison with manual TA confirmed that most themes were consistent. Given the limited size of our dataset, the implications of this study should be regarded as preliminary. Nevertheless, our findings suggest that NLP may serve as a useful exploratory approach to complement manual TA when analyzing larger free-text datasets on medical needs in future

**Data availability statement:** The dataset generated in this study includes free-text responses, and complete removal of potentially identifiable information cannot be fully guaranteed. Therefore, there are ethical restrictions on publicly sharing the dataset. These restrictions have been imposed by the Ethical Committee, Faculty of Medicine, The University of Tokyo. Although the full dataset cannot be made publicly available, we are able to provide a minimal anonymized dataset upon reasonable request, in accordance with the ethical guidelines. Data requests may be sent to the 22q Research Office at the University of Tokyo (22q.research@gmail.com), which serves as the designated contact point for data access inquiries.

**Funding:** This study was partially supported by Japan Society for the Promotion of Science (JSPS) KAKENHI (Grant Numbers JP21H05171 to KK & AK, JP21H05174 to KK & AK, JP21H00451 to KK, JP21K13474 to AK, and JP23K27525 to KK), the Research and Development Grants for Comprehensive Research for Persons with Disabilities from the Japan Agency for Medical Research and Development (AMED) (Grant Number: 20ek0109369 to KK), the AMED Multidisciplinary Frontier Brain and Neuroscience Discoveries (Brain/MINDS 2.0) (Grant Number: JP23wm0625001 to KK), the AMED Program for Research Ethics and Social Co-creation (Grant Number: JP25oa0439005h0001 to KK), the International Research Center for Neurointelligence (WPI-IRCN) at The University of Tokyo Institutes for Advanced Study (UTIAS) to KK, and the Yuumi Memorial Foundation for Home Health Care to YS. The funders had no role in study design, data collection and analysis, decision to publish, or preparation of the manuscript.

**Competing interests:** The authors have declared that no competing interests exist.

research. NLP should not be viewed as a replacement for manual TA, but rather as a supportive method that offers additional perspectives and potential patterns.

## Introduction

22q11.2 deletion syndrome (22q11DS) is a chromosomal disorder caused by a microdeletion in chromosome 22, with more than 90% of cases occurring *de novo* [1]. The morphological and functional impairments across multiple organs and systems due to the deletion result in disabilities across physical, intellectual, and psychiatric domains. These disabilities evolve and accumulate throughout different life stages [2,3]. For instance, newborns may require repeated hospitalizations due to congenital heart disease, school-age children may struggle academically due to intellectual disabilities or borderline intelligence, and adolescents may experience mental health issues [4,5]. Due to the overlap of these conditions, there is often a mismatch in healthcare frameworks, leading to unmet support needs.

However, few studies have investigated medical challenges and unmet needs in individuals with 22q11DS [6], and little is known about how patients and their family perceive these issues. Qualitative research has gained attention for capturing subjective indicators and narratives of patients [7]. Among qualitative research methodologies, thematic analysis (TA) is particularly valued for identifying and interpreting themes across entire datasets [8].

On the other hand, some researchers acknowledge that bias is inevitable in qualitative research [9]. Recent research has evaluated the use of large language models in TA, critically examining potential human researchers' biases. Two studies applied ChatGPT-based TA, indicating that the manual TA and natural language processing (NLP)-based thematic extraction were largely consistent [10,11]. They suggest that large language models can serve as an additional member of the analysis team by contributing novel perspectives and offering alternative interpretations of the identified codes.

We adopted a critical realist perspective, which assumes that complex phenomena cannot be fully captured through a single analytic method and are better understood by integrating multiple methodological lenses [12]. Accordingly, we implemented method triangulation [13] by combining human-led TA with NLP-based automated category extraction. This approach allows us to complement potential heuristics and biases that may arise during researcher-driven coding, while providing a more comprehensive and structured perspective on the diverse and complex needs characteristic of individuals with 22q11DS. These additions strengthen the theoretical coherence of the study and articulate the methodological gap that the combined qualitative–computational approach aims to address.In the design of this study, NLP was incorporated not as a replacement for manual TA but as an exploratory, complementary method. Manual TA was first used to identify core themes, after which NLP was applied to explore additional perspectives and to generate a more structured thematic representation. This design allowed us to integrate insights from both approaches to achieve a broader understanding of caregivers' descriptions.

In terms of evaluation, our goal was not to assess perfect concordance between manual TA and NLP. Instead, we adopted an evaluative framework that examines how the two methods jointly contribute complementary themes and illuminate different aspects of the data. This approach reflects our assumption that multiple analytic methods can reveal different layers of complex phenomena, consistent with our critical realist stance.

In this study, we aimed to unveil medical challenges and unmet needs collected through a web-based survey, using TA. We also developed an automated thematic extraction workflow using NLP, anticipating the analysis of larger-scale qualitative data in future research.

## Methods

### Ethical considerations

Ethical considerations were addressed following the Declaration of Helsinki, with approval from the Ethics Committee of the University of Tokyo Graduate School of Medicine and Faculty of Medicine [approval no. 2018015NI]. This study was conducted using a web-based survey. On the first page of the survey, participants were provided with written information regarding the study purpose, the voluntary nature of participation, the anonymity of the data collected, and their right to withdraw at any time. Participants were instructed to carefully review this information before proceeding. In accordance with the approval granted by the institutional ethics committee, informed consent was obtained through the participants' initiation of the survey, whereby the submission of responses was considered to indicate informed consent.

All participants in this study were adults and primary caregivers; therefore, no minors were included, and consent from parents or legal guardians was not required. Participants were instructed about the purpose of this study on the first page of the web-based survey. We also explained that informed consent was obtained through the participant's response to the survey.

### Target population

125 primary caregivers (mothers or fathers) of individuals with 22q11DS participated in a web-based survey conducted between March 20 and November 8, 2019. A response to this survey was restricted to once per family. The survey was disseminated via email and postal mail with the support of the 22q11DS family association (22 HEART CLUB), the family associations of patients with pediatric heart disease, and medical staff involved in the care of individuals with 22q11DS. For participants who found it difficult to answer the web-based survey, the paper questionnaire was sent by mail upon request. Four out of 125 valid responses were via the paper questionnaire. The development process of the survey questions has been described in detail in our prior work [6,14].

Of the 125 participants, we analyzed responses from 65 caregivers who answered the open-ended question, "What kinds of support do you think are needed for individuals with 22q11DS and their parents in the context of medical care?" Descriptive statistics of respondent characteristics are shown in Table 1 (n = 65).

### Thematic analysis (manual)

We conducted TA on the free-text responses (n = 65) concerning medical challenges and needs collected through a web-based survey, referring partially to Braun & Clarke (2006) [15].

To enhance rigor, two researchers independently generated initial codes from a randomly selected subset of 25% of the responses. Their coding notes were then compared, and similar codes were merged and organized into a hierarchical structure (initial codes, subcategories, categories). The definitions of individual codes were iteratively refined to reduce redundancy and ambiguity, resulting in clear operational definitions. Relationships among subcategories were examined to structure them toward higher-order categories.

**Table 1. Descriptive statistics of study participants (total n = 65).**

| Caregiver | | | | |
|---|---|---|---|---|
| | Age, mean (sd) | | 45.4 | (7.3) |
| | Mother, num (%) | | 57 | (87.7) |
| | Father, num (%) | | 8 | (12.3) |
| | Annual household income, num (%) | | | |
| | | 0-2.99 million yen | 6 | (9.2) |
| | | 3-5.99 million yen | 19 | (29.2) |
| | | 6-8.99 million yen | 23 | (35.4) |
| | | 9-11.99 million yen | 4 | (6.2) |
| | | 12-14.99 million yen | 8 | (12.3) |
| | | 15 million yen or over | 5 | (7.7) |
| | Marital status, num (%) | | | |
| | | Single, Divorced, Widowed | 8 | (12.3) |
| | | Married | 57 | (87.7) |
| | K6 score, mean (sd) | | 5.2 | (5.1) |
| | | | | |
| **Person with 22q11.2 deletion syndrome** | | | | |
| | Age, mean (sd) | | 12.1 | (8.2) |
| | Women or girl, num (%) | | 33 | (50.8) |
| | | | | |
| Lifetime comorbidities, mean (SD) | | | 4.7 | (1.7) |
| Lifetime comorbidities, N (%) | | | | |
| | Congenital heart disease | | 57 | (87.7) |
| | Immune system disorder | | 23 | (35.4) |
| | Endocrine disorder | | 27 | (41.5) |
| | Gastrointestinal disease | | 22 | (33.8) |
| | Otorhinolaryngology/maxillofacial disease | | 50 | (76.9) |
| | Orthopedic disease | | 30 | (46.2) |
| | Growth/developmental disorder | | 58 | (89.2) |
| | Psychiatric/neurological disorder | | 24 | (36.9) |
| | Other | | 15 | (23.1) |

After independent coding, discrepancies between the two coders were identified and discussed in depth, and consensus was reached through iterative discussion. The intercoder agreement rate (90%) was calculated using simple percentage agreement. Once consensus on the coding framework was established, the first author proceeded to code the remaining data.

Reflexivity was maintained throughout the analysis. The two coders brought distinct forms of expertise and positionality: the first author is a psychiatrist working in a specialized outpatient clinic for individuals with 22q11DS and is a beginner in TA, whereas the second author is a clinical psychologist with substantial qualitative research experience and close engagement with participants through recruitment, communication, and follow-up. Analytic memos were shared to reflect on how these backgrounds might influence interpretations, and potential biases were discussed and adjusted during the refinement of categories and themes. This iterative process ensured that interpretations did not become overly shaped by a single researcher's standpoint.

### Natural language processing-based thematic extraction

To protect personal information, all analyses were conducted in a fully offline environment. Inference was performed using two NVIDIA RTX 6000 Ada GPUs (48 GB each). We selected a quantized model that could be loaded entirely into the available GPU memory to enable secure, local execution. Quantization refers to reducing the numerical precision of model weights (e.g., from 16-bit to 4- or 8-bit), which substantially lowers memory and computational requirements while maintaining sufficient performance for tasks such as thematic extraction. We used a quantized version of the open-source large language model CohereForAI/c4ai-command-r-plus [16], specifically bartowski/c4ai-command-r-plus-08–2024-GGUF (Q6_K) [17], which provided the highest precision compatible with our hardware constraints. Q6_K corresponds to a 6-bit quantization scheme in the GGUF format. No additional training or domain-specific fine-tuning was performed.

In the NLP-based thematic extraction, we did not employ external algorithms such as embeddings or clustering. Instead, we used a prompt-engineering approach in which the model followed a stepwise set of analytic procedures. This process was carried out without reference to the results of the manual TA. A system prompt instructed the model to act as a qualitative research scholar, and subsequent prompts directed it through sequential stages analogous to manual TA, including familiarization with the data, generation of initial codes, comparison and integration of related codes, and the derivation and naming of themes. The prompts also incorporated a tree-of-thoughts structure in which two simulated researchers (A and B) independently proposed interpretations and reached consensus through comparison, as well as iterative refinement using mock outputs [18]. This structured prompting allowed the model to execute analytic steps corresponding to the hierarchical stages of manual TA (initial codes, categories, themes). Because the full prompts are lengthy and include detailed stepwise instructions, they are provided in the Supplementary Information (S1 File).

All theme and subcategory labels reported in this study were generated automatically by the NLP model without any rephrasing or merging by the researchers. No human-in-the-loop editing was applied during the theme generation process.

## Results

### Manual thematic analysis

We extracted 87 labels from the responses analysed. We established Category 1 consisting of two subcategories (a – b), Category 2 consisting of three subcategories (a – c), Category 3 consisting of two subcategory (a – b), Category 4 consisting of three subcategories (a – c), and Category 5 consisting of one subcategory (a) (Table 2).

In Category 1, "Comprehensiveness", we identified requests for the establishment of comprehensive support services, with emphasis on the difficulty of accessing comprehensive medical support due to overlapping medical conditions. In Category 2, "Lack of Knowledge and Information", caregivers expressed concerns about the uncertainty of future outcomes and highlighted the need for access to professionals with specialized knowledge, as well as support in gathering reliable information. In Category 3, "Individuality", there were calls for greater recognition of individual differences, including the need to accommodate sensory hypersensitivity in medical and support settings. In Category 4, "Consideration for Mental Well-being", participants emphasized the importance of responses that support both the mental well-being of the child and the parents, noting that the attitudes of those in close proximity significantly affect the family's sense of security. In Category 5, "Positive Experiences", the value of sharing concrete, positive examples of successful support was noted as a way to encourage and inform other families in similar situations.

The themes and quoted free-text responses presented in this study were translated from Japanese into English by the first author. Although back-translation was not performed, the translations were reviewed by multiple healthcare professionals with clinical experience in the care of individuals with 22q11DS to ensure accuracy and preservation of medical context and implied meanings.

**Table 2. Qualitative analysis of free-text responses to the questions regarding medical needs.**

| Category<br> Subcategory | Excerpt from narrative |
|---|---|
| **1. Need for Comprehensive Support**<br>a. The establishment of comprehensive support services is necessary. | Support that provides advice from care managers or coordinators on how to respond in case of an emergency.<br>Because symptoms vary from person to person, it would be helpful to have support that offers advice when difficulties arise in daily life. |
| b. Due to overlapping medical conditions, access to comprehensive medical support is challenging. | In cases where multiple conditions overlap, there should be a department that provides integrated medical care in collaboration with other specialties and hospitals. |
| **2. Lack of Knowledge and Information**<br>a. Difficulty in predicting future outcomes. | We would like to receive prior explanations about symptoms or difficulties that may emerge with aging. |
| b. The presence of staff with specialized knowledge is essential. | In some regions, there are no specialists available. |
| c. Support for information gathering is needed. | Support is needed for those who do not know how to access the information they want. |
| **3. Understanding Individual Differences**<br>a. Consideration for sensory hypersensitivity is required. | Due to sensory hypersensitivity, our child expressed to us that hearing other children in the same room was more distressing than post-surgical pain. |
| b. Recognition of individual differences is important. | Since there are significant individual differences, we want others to understand and acknowledge them. |
| **4. Consideration for Mental Well-being**<br>a. Responses should take mental well-being into account. | We would like every department to have doctors who not only focus on physical health but also consistently care for the patient's mental well-being. |
| b. A support system for parents should be in place. | Counseling for parents regarding the child's psychological or emotional issues. |
| c. The attitudes of people in close proximity play a significant role. | |
| **5. Sharing of Positive Experiences**<br> a. Sharing concrete success stories is valuable. | We appreciate that our regular pediatrician provides advice to parents. |

## Thematic extraction via NLP

The NLP-based method extracted established Category 1 consisting of two subcategories (a – b), Category 2 consisting of two subcategories (a – b), and Category 3 consisting of two subcategories (a – b) (Table 3).

In Category 1, "Child-Centered Approach", emphasis was placed on the need for care and support tailored to children's health conditions and developmental stages, along with calls for mental health and psychological support that responds to individual needs. In Category 2, "Comprehensive Support Across Healthcare and Welfare Services", participants highlighted the importance of comprehensive support and coordination by professionals, as well as the implementation of early-stage risk management and intervention strategies. In Category 3, "Parental Support Through Social and Community Networks", there were requests for mechanisms to reduce caregiver isolation through information sharing and accessible consultation services, along with educational support to improve understanding of child-rearing and developmental issues.

Although the model produced response IDs that it associated with each theme, the accuracy of assigning individual responses to specific subcategories was insufficient; many responses were disproportionately concentrated in a single

**Table 3. Categories extracted using natural language processing.**

| Category<br>  Subcategory |
| --- |
| **1. Child-Centered Approach** |
| a. Emphasizes the importance of care and support tailored to children's health and developmental stages. |
| b. Addresses the need for mental health and psychological support. |
| **2. Comprehensive Support Across Healthcare and Welfare Services** |
| a. Highlights the necessity for comprehensive medical care and coordination provided by experts. |
| b. Advocates for the implementation of risk management and early intervention strategies from an early stage. |
| **3. Parental Support Through Social and Community Networks** |
| a. Supports the reduction of loneliness through information sharing and the establishment of consultation services. |
| b. Educational support to enhance understanding of child care and development. |

subcategory. Therefore, these automatically generated representative IDs were removed from the final presentation of the NLP results (Table 3).

To improve transparency, we included a brief description of the prompts used for theme generation in this section and provided examples of the raw model outputs in the Supplement (S2 File).

## Discussion

A key methodological contribution of this study lies in demonstrating the feasibility of conducting NLP-assisted qualitative analysis within a fully offline, privacy-preserving environment. Previous applications of large language models (LLMs) for TA relied on cloud-based systems [10,11], which posed challenges when handling sensitive personal narratives [19]. In this study, we developed an analysis workflow that employs an open-source LLM executed entirely on local hardware, enabling secure processing of confidential qualitative data (Table 4). This approach provides a practical framework for researchers working with protected health information while also possessing characteristics that make it potentially applicable to participatory research. Recent work has shown that NLP can extract major themes from large volumes of free-text data and complement traditional qualitative approaches [20]. Studies integrating NLP with TA further suggest that preliminary computational themes can serve as a useful starting point for collaborative interpretation [21], indicating that NLP may support co-analysis in participatory research settings. Moreover, participant trust, psychological safety, and robust data protection are recognized prerequisites for the use of digital technologies in healthcare [22], and secure,

**Table 4. Comparison between prior ChatGPT-based thematic analysis studies and the present NLP-assisted approach.**

| Aspect | Prior studies (ChatGPT-based TA) | Present study |
| --- | --- | --- |
| Model used | ChatGPT (commercial LLM) | Open-source LLM |
| Execution environment | Cloud-based | Local (on-premises) |
| Data transmission | Sent to external servers | No external data transmission (fully offline) |
| Privacy protection | Dependent on service provider policies | Fully controlled by researchers |
| Ethical considerations | Limited or not explicitly described | Explicitly designed to avoid external data transfer |

locally executed workflows can help create conditions in which participants feel comfortable sharing their experiences. Taken together, our workflow—which combines secure data handling with NLP-assisted thematic exploration—may provide a methodological foundation that supports participatory research involving collaborative analysis among patients, caregivers, and healthcare professionals.

In line with a critical realist perspective, we assumed that the complex medical needs of individuals with 22q11DS could not be fully captured through a single analytic method. Therefore, we employed manual TA and NLP-based thematic extraction in a complementary manner. By using NLP to illuminate themes that might not be fully identified through manual analysis alone, our aim was to obtain a more comprehensive and expansive understanding of the challenges and unmet needs described by caregivers.

We interpret the findings by comparing the results of the manual TA and the NLP-based thematic extraction. We created a mapping table (Table 5) that summarizes the overlaps and divergences between subcategories. The differences observed between the two approaches reflect their distinct analytic characteristics. Manual TA enabled the generation of themes informed by researchers' clinical experience and interpretive judgment, resulting in themes that were identified only through human analysis. Conversely, the NLP approach identified themes that were less salient from a human interpretive perspective, thereby complementing the manual analysis. For interpretive clarity, we retrospectively characterized the themes identified by manual TA and NLP in terms of their relative level of abstraction. Micro-level themes refer to concrete and specific needs directly grounded in caregivers' descriptions, corresponding to subcategories identified by both approaches. Meso-level themes integrate multiple micro-level themes into coherent domains and largely reflect the categories generated through manual TA. In contrast, macro-level themes represent broader and more abstract constructs that aggregate multiple themes and were more characteristic of the categories generated by NLP. When comparing the two approaches, concordance was most evident at the micro-level, while differences at the meso- and macro-levels reflect the distinct analytic properties of human interpretation and computational abstraction.

Some of the higher-level categories generated by the NLP model include abstract concepts that do not appear verbatim in the original free-text responses. However, these categories were derived by integrating recurring semantic patterns across multiple responses, such as the need for ongoing guidance in parent–child relationships, and expectations for support from others. We therefore interpret these themes not as model hallucinations, but as instances of conceptual synthesis through abstraction. At the same time, large language models are known to have a tendency toward overgeneralization and the production of normative language. Accordingly, NLP-generated higher-level themes should be interpreted cautiously, with careful attention to their faithfulness to the underlying responses [23].

Because all free-text responses analyzed in this study were written by caregivers, the findings reflect caregivers' perceptions of medical challenges and unmet needs rather than direct reports from individuals with 22q11DS themselves. These descriptions include both the needs that caregivers perceive as necessary for the individuals and the needs they personally experience in the course of accompanying medical visits or providing daily support. This distinction is important for interpreting the results, as the themes represent caregiver-recognized challenges and unmet needs, which may not fully align with the individuals' own perspectives.

Although this study analyzed only the free-text responses written in the "medical care" section of the questionnaire, some caregivers described issues that spanned both medical and welfare domains. In practice, medical challenges and welfare support needs are often closely intertwined in the daily lives of families, making them difficult to distinguish (e.g., *"It would be helpful to have a centralized consultation point like a general medicine department and a coordinator such as a care manager or social worker."*). Consequently, some responses in the medical section included welfare-related content, such as the need for a centralized consultation point or coordination by a care manager or social worker. Because the NLP-based thematic extraction reflects the vocabulary and contextual information provided by respondents, welfare-related elements were also captured as themes. This overlap should therefore be understood as a characteristic of the responses themselves rather than a limitation of the NLP approach.

**Table 5. Correspondence between manual thematic analysis and natural language processing-based thematic extraction subcategories.**

| Manual TA Subcategory | NLP-based Subcategory | Exemplar quotations from manual TA |
|---|---|---|
| 1a. The establishment of a comprehensive support service is necessary. | 2a. Highlights the necessity for comprehensive medical care and coordination. | "It would be helpful to have a centralized consultation point like a general medicine department and a coordinator such as a care manager or social worker." |
| 1b. Due to overlapping medical conditions, access to comprehensive medical support is challenging. | 2b. Advocates for the implementation of risk management and early intervention strategies from an early stage. | "Because my child's condition changes from time to time, I would like to have access to a specialized consultation service." |
| 2a. Difficulty in predicting future outcomes. | 1a. Emphasizes the importance of care and support tailored to children's health and developmental stages. | "We would like to receive prior explanations about symptoms or difficulties that may emerge with aging." |
| 2b. The presence of staff with specialized knowledge is essential. | 3b. Educational support to enhance understanding of child care and development. | "I thought it would be helpful to have something like a pass card that contains the medical information and explanations necessary for my child." |
| 2c. Support for information gathering is needed. | 3b. Educational support to enhance understanding of child care and development. | "Support is needed for those who do not know how to access the information they want." |
| 4a. Responses should take mental well-being into account. | 1b. Addresses the need for mental health and psychological support. | "We would like every department to have doctors who not only focus on physical health but also consistently care for the patient's mental well-being." |
| 4b. A support system for parents should be in place. | 3b. Educational support to enhance understanding of child care and development. | "We need someone who can give us ongoing, practical guidance on how to build a positive parent–child relationship without increasing our worries." |
| 3a. Consideration for sensory hypersensitivity is required. | — | **Manual only** |
| 3b. Recognition of individual differences is important. | — | **Manual only** |
| 4c. The attitudes of people in close proximity play a significant role. | — | **Manual only** |
| 5a. Sharing concrete success stories is valuable. | — | **Manual only** |
| — | 3a. Supports the reduction of loneliness through information sharing and consultation services. | **NLP only** |

This table presents the correspondence between subcategories identified through manual thematic analysis (TA) and those generated by the natural language processing (NLP)–based thematic extraction. Subcategories listed in the left column were derived from manual TA, while those in the right column were produced through the NLP workflow. When both methods identified conceptually comparable themes, the corresponding subcategories are presented in the same row; entries marked with "—" indicate subcategories that were unique to one method. NLP-only and TA-only themes are listed at the bottom of the table. For subcategories that were identified by both manual TA and the NLP-based extraction, exemplar quotations are provided. To ensure accuracy and interpretability, all exemplar quotations were selected by the researchers based solely on the manual TA of the original free-text responses.

Manual TA and NLP-based thematic extraction identified common themes on medical challenges and needs of individuals with 22q11DS. One theme was extracted only by NLP method: reducing caregivers' loneliness through information sharing and consultation services (Subcategory 3a identified via NLP). Previous research indicated that approximately half of caregivers experience feelings of loneliness [24], underscoring the critical role of healthcare professionals in deepening their understanding of the challenges faced by families and in contributing to the psychological well-being of caregivers [25].

Some themes were extracted only by manual TA: understanding individual differences (Subcategory 3a and 3b identified via manual TA), the attitudes of people in close proximity (Subcategory 4c identified via manual TA), and sharing of positive experiences (Subcategory 5a identified via manual TA). The reason why these themes were identified only through manual TA may be that the researchers' positionality, shaped by our clinical experience [26] —such as an emphasis on the need to consider individual differences (e.g., sensory sensitivities) or the value of making positive perspectives more visible.

Common themes identified by manual TA and NLP align with the complexity and evolving nature of medical needs in individuals with 22q11DS. Because the syndrome affects multiple organ systems, comprehensive medical management is crucial, requiring coordinated care across multiple specialties, including cardiology, immunology, endocrinology, neurology, psychiatry, and developmental medicine [2, 3] (Subcategory 1a and 1b identified via manual TA; Subcategory 2a and 2b identified via NLP). Furthermore, medical challenges shift with age, from congenital heart defects and feeding difficulties in infancy to learning disabilities and psychiatric conditions in adolescence and adulthood (Subcategory 2a identified via manual TA; Subcategory 1a identified via NLP). The need for structured, age-specific interventions and multidisciplinary collaboration underscores the importance of early diagnosis and continuous follow-up [27]. Additionally, the high prevalence of psychiatric disorders in adulthood, such as schizophrenia, highlights the necessity of long-term mental health support [4, 5, 28] (Subcategory 4a identified via manual TA; Subcategory 1b identified via NLP). Improving access to specialized care and addressing caregivers' concerns through systematic interventions are critical in enhancing health outcomes and quality of life for individuals with 22q11DS (Subcategory 2b, 2c, and 4b identified via manual TA; Subcategory 3b identified via NLP).

## Limitations

First, there is a possibility of sampling bias, although the survey was distributed through family associations and websites across Japan. Because our group operates the only specialized psychiatric outpatient clinic for 22q11DS in Japan, caregivers of individuals with more severe difficulties and needs may have been more likely to participate. There also may be sampling bias between respondents (Table 1) and non-respondents (S1 Table) to the free-text question on medical needs. Although no substantial differences were observed in mean age or sex distribution, respondents exhibited a greater variety of physical comorbidities, with slightly higher prevalence of conditions such as congenital heart disease, immune system disorders, and endocrine disorders. In addition, the prevalence of psychiatric disorders among respondents was approximately twice that observed among non-respondents (36.9% vs. 18.3%). These patterns suggest that families experiencing more complex medical and psychiatric challenges may have been more likely to provide free-text responses, and thus the results of this study may disproportionately reflect the perspectives of individuals with more complex healthcare needs. Second, while the NLP-based analysis was effective in extracting themes, it demonstrated limited capacity in accurately assigning individual responses to those themes. Consequently, we were unable to quantitatively compare the two approaches, such as calculating concordance rates between NLP-based classification and manual TA. Third, this study collected responses solely from family members, and not from the individuals with 22q11DS themselves. In future research, we aim to incorporate methods that facilitate participation by individuals with 22q11DS, such as utilizing visual aids (e.g., illustrations or photographs) with the support of caregivers or staff. Fourth, NLP outputs may be substantially influenced by prompt design. The prompts used in this study could have shaped the thematic extraction process

and the resulting category structure, and different prompts or model configurations may yield different results. Future research should compare multiple prompt and model conditions to examine the robustness of NLP-derived findings. Fifth, the free-text responses analyzed in this study were generated by caregivers in Japan, and thus may reflect cultural and institutional factors that shape narrative style and the expression of needs. Given the influence of the Japanese health-care system, family norms, and caregiving context, generalizability to other languages or healthcare systems should be approached with caution. Sixth, LLMs may incorporate cultural and linguistic biases derived from their training data and model design [29]. In particular, free-text responses written by Japanese caregivers often include idiomatic expressions and implicit meanings, which LLMs may misinterpret or over-generalize during abstraction. Seventh, NLP-based thematic extraction is inherently dependent on prompt design, and even minor variations in prompts may influence the resulting outputs. In addition, when summarizing and integrating multiple narratives, LLMs may generate synthetic or normative phrasing that does not appear verbatim in the original text, raising concerns about faithfulness to the source data [23]. Eighth, while NLP-based analysis is useful for obtaining an overview of general trends and shared patterns across large volumes of qualitative data, it is less suited to capturing emotional intensity and subtle contextual nuances embedded in individual narratives [30].

## Conclusion

The primary contribution of this study is methodological. We show that NLP-assisted qualitative analysis using LLMs can be conducted within a fully offline, privacy-preserving environment in healthcare research. This workflow offers a practical approach for applying NLP to sensitive qualitative data while maintaining data security.

The thematic findings are presented as an illustrative application of this methodological approach. The NLP-based analysis identified themes—such as the need for comprehensive support systems for medical consultation and attention to mental health—that were consistent with those derived from manual TA. Together, these results suggest that NLP would complement, rather than replace, human-led qualitative analysis in medical research.

## Supporting information

**S1 File. English translations and materials used for prompt engineering procedures.** This file includes the English translations of the step-by-step prompts, mock outputs, and revised prompts used in the prompt engineering procedures applied in this study.
(DOCX)

**S2 File. English translations of raw outputs generated by the NLP model, in which the model simulated two independent analytic perspectives ("Researcher A" and "Researcher B") and compared them to derive thematic interpretations.** These examples are provided to illustrate the model's internal reasoning process.
(DOCX)

**S1 Table. Clinical characteristics of participants who did not provide free-text responses in the medical needs section (n = 60).**
(DOCX)

## Author contributions

**Conceptualization:** Yutaka Sawai, Miho Tanaka, Sho Yaghishita, Kiyoto Kasai.

**Data curation:** Yutaka Sawai, Miho Tanaka, Kiyoto Kasai.

**Formal analysis:** Yutaka Sawai, Miho Tanaka, Kiyoto Kasai.

**Funding acquisition:** Yutaka Sawai, Sho Yaghishita, Kiyoto Kasai.

**Investigation:** Yutaka Sawai, Miho Tanaka, Shimon Tonsho, Akito Uno, Yusuke Takahashi, Akiko Kanehara, Yousuke Kumakura, Sho Yaghishita, Kiyoto Kasai.

**Methodology:** Yutaka Sawai, Miho Tanaka, Shimon Tonsho, Sho Yaghishita, Kiyoto Kasai.

**Project administration:** Yutaka Sawai, Miho Tanaka, Sho Yaghishita, Kiyoto Kasai.

**Resources:** Yutaka Sawai, Miho Tanaka, Akiko Kanehara, Sho Yaghishita, Kiyoto Kasai.

**Software:** Yutaka Sawai, Miho Tanaka, Shimon Tonsho, Sho Yaghishita, Kiyoto Kasai.

**Supervision:** Akiko Kanehara, Yousuke Kumakura, Sho Yaghishita, Kiyoto Kasai.

**Validation:** Yutaka Sawai, Miho Tanaka, Shimon Tonsho, Kiyoto Kasai.

**Visualization:** Yutaka Sawai, Miho Tanaka.

**Writing – original draft:** Yutaka Sawai.

**Writing – review & editing:** Yutaka Sawai, Miho Tanaka, Shimon Tonsho, Akito Uno, Yusuke Takahashi, Akiko Kanehara, Yousuke Kumakura, Sho Yaghishita, Kiyoto Kasai.

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
