## [Decision Letter · Decision Letter 0]

17 Nov 2025

PMEN-D-25-00414

Medical challenges and unmet needs of individuals with 22q11.2 deletion syndrome: A thematic analysis and natural language processing-based thematic extraction

PLOS Mental Health

Dear Dr. Kasai,

Thank you for submitting your manuscript to PLOS Mental Health. After careful consideration, we feel that it has merit but does not fully meet PLOS Mental Health's publication criteria as it currently stands. Therefore, we invite you to submit a revised version of the manuscript that addresses the points raised during the review process.

• A rebuttal letter that responds to each point raised by the editor and reviewer(s). You should upload this letter as a separate file labeled 'Response to Reviewers'.

We look forward to receiving your revised manuscript.

Kind regards,

Dr. Wenwang Rao

Academic Editor

PLOS Mental Health

Journal Requirements:

1. Please provide additional details regarding participant consent. In the ethics statement in the Methods and online submission information, please ensure that you have specified (1) whether consent was informed and (2) what type you obtained (for instance, written or verbal, and if verbal, how it was documented and witnessed). If your study included minors, state whether you obtained consent from parents or guardians. If the need for consent was waived by the ethics committee, please include this information.

i. State what role the funders took in the study. If the funders had no role in your study, please state: “The funders had no role in study design, data collection and analysis, decision to publish, or preparation of the manuscript.”

3. Please ensure that your Ethics Statement is available in its entirety at the beginning of your Methods section, under a subheading 'Ethics Statement'.

4. We have noticed that you have cited Table 2 in the manuscript file but there are no corresponding tables in the manuscript. Please amend your manuscript to include this table, noting that tables should not be uploaded as individual files.

5. We have noticed that you have uploaded Supporting Information files, but you have not included a list of legends. Please add a full list of legends for your Supporting Information files after the references list.

6. We note that you have indicated that there are restrictions to data sharing for this study. For studies involving human research participant data or other sensitive data, we encourage authors to share de-identified or anonymized data. However, when data cannot be publicly shared for ethical reasons, we allow authors to make their data sets available upon request. For information on unacceptable data access restrictions, please see http://journals.plos.org/plosone/s/data-availability#loc-unacceptable-data-access-restrictions

Additional Editor Comments (if provided):

None

Reviewers' comments:

Reviewer's Responses to Questions

**Comments to the Author**

1. Does this manuscript meet PLOS Mental Health’s publication criteria ? Is the manuscript technically sound, and do the data support the conclusions? The manuscript must describe methodologically and ethically rigorous research with conclusions that are appropriately drawn based on the data presented.

Reviewer #1: Yes

Reviewer #2: Yes

Reviewer #3: Partly

2. Has the statistical analysis been performed appropriately and rigorously?

Reviewer #1: Yes

Reviewer #2: Yes

Reviewer #3: I don't know

3. Have the authors made all data underlying the findings in their manuscript fully available (please refer to the Data Availability Statement at the start of the manuscript PDF file)?

Reviewer #1: Yes

Reviewer #2: Yes

Reviewer #3: No

4. Is the manuscript presented in an intelligible fashion and written in standard English?

Reviewer #1: Yes

Reviewer #2: Yes

Reviewer #3: Yes

Reviewer #1: To Authors:

The manuscript addresses an important and underexplored topic concerning the medical and psychosocial needs of individuals with 22q11.2 deletion syndrome (22q11DS) and their caregivers. The dual application of traditional thematic analysis (TA) and natural language processing (NLP)-based thematic extraction is timely and potentially valuable.

Comment 1: The research aim and rationale require clearer logical coherence. Specifically, the manuscript should more explicitly articulate how the NLP-based analysis advances existing qualitative approaches, beyond efficiency and scalability. A sharper statement of the theoretical and methodological gap that the study fills would strengthen the Introduction.

Comment 2: The methodological rigor and transparency are generally adequate, but several details need clarification for reproducibility. The authors should describe the exact prompts used for the NLP analysis (or summarize them in the main text rather than relegating all to the supplement) and specify how the “themes” were algorithmically defined and aggregated. For the manual TA, the coding framework and the process for category refinement should be explained more concretely, including how discrepancies were resolved and how reflexivity was maintained among coders.

Comment 3: While the use of an open-source LLM in a secure offline environment is commendable, the technical explanation could be made more accessible to readers in health sciences. Briefly explain what distinguishes the “quantized” model from the original and why this configuration was chosen. Readers unfamiliar with machine learning may find it difficult to follow the computational logic without some interpretive guidance.

Comment 4: The comparative validity between the manual and NLP-based analyses is discussed qualitatively but would benefit from quantitative or structured evaluation. Even if statistical concordance is not possible, presenting an explicit mapping table showing which categories overlapped, diverged, or were newly generated (with frequency counts or exemplar quotations) would enhance credibility and reproducibility.

Comment 5: The Results section is well organized, but the tables could be improved by integrating key quotes from caregivers that exemplify the derived subcategories. Doing so would make the findings more vivid and support claims about interpretive alignment between the two methods.

Comment 6: In the Discussion, the integration of findings with prior literature is appropriate, yet parts of the argument overstate the implications. The claim that “NLP could be useful to analyze large-scale free-text data on medical needs” should be qualified as preliminary given the limited sample size (n = 65 free-text responses). The authors might more cautiously frame NLP as a complementary exploratory tool rather than a replacement for manual analysis.

Comment 7: The Limitations section is appreciated, but additional methodological caveats should be acknowledged—particularly the potential influence of prompt design on NLP outputs and the cultural context of Japanese caregiver narratives, which may constrain generalizability to other languages or health systems.

Comment 8: Regarding novelty and contribution, the study’s strength lies in demonstrating the feasibility of privacy-preserving NLP-assisted qualitative analysis within medical research. To make this contribution more visible, the authors could reorganize the Discussion around this methodological innovation and discuss its potential integration into participatory or mixed-methods research frameworks.

Reviewer #2: Comments to Authors

Sawai et al. reported the results of a qualitative study on the medical issues and unmet needs of individuals of 22q11.2 deletion syndrome in Japan. The study used data from a web-based survey and conducted a thematic analysis using manual coding and NLP methods. The authors highlighted the importance of comprehensive improvements in the healthcare system and support for caregivers. The authors also developed an automated thematic extraction workflow using NLP for better analysis of larger-scale qualitative data.

This study will contribute to clarifying the medical support needed by individuals with multiple conditions and their caregivers. However, there are several points that need to be addressed to further improve the manuscript.

Title

1. Since this study is based on free-text comments provided by caregivers, the authors need to consider which wording of the title is appropriate to describe this study; Medical challenges and unmet needs of individuals themselves or caregivers.

Methods

2. The demographics of the participants is not listed in the manuscript. Although the demographics of the original 125 participants of the survey can be found in their previous papers, cited as ref. 6 and 12, the demographics of the 65 participants provided free-text responses should be presented.

3. The authors should describe the procedure used to extract comments in this study. For example, how many of the 125 participants provided free-text comments, and among them, how many commented specifically on medical issues? In addition, were there any notable characteristics or trends between those who provided free-text comments and those who did not? If there were differences between 2 groups in factors such as the severity of the physical or mental condition of individuals or the participants’ regions of residence (e.g. urban and rural areas regarding medical access), the authors should discuss them. There may be distinctive motivations that encourage caregivers to leave comments.

4. Regarding manual thematic analysis, the authors described “1. Two researchers independently coded responses from a randomly 89 selected subset of 16 participants (25%). 2. After confirming an inter-coder agreement rate of over 90%, the first author coded the remaining responses.”

Please specify whether the two researchers who manually coded the text in this study included the first author or were two independent individuals. Also, it is better to describe briefly whether there were any differences in their experience or expertise related to thematic analysis.

Results

5. The aim of this study appears to be to identify medical issues. Actually, in their previous work cited as ref. 6, the authors also divided discussions into medical, welfare, and educational domains. However, in the present results, particularly presented in table 2 (provided via NLP), it seems that contents related to the “welfare” domain are included. For example, category 2 contains the term “welfare services,” and category 3 includes “Social and Community Networks.”

The authors should clarify whether the target of this study is “medical” issues only, or both “medical and welfare” issues. Alternatively, if this overlap is due to technical limitations or characteristics of the NLP program they developed, it should be discussed.

6. Another concern regarding table 2 is that no excerpt is provided in cell 2a. Is there any reason to left this cell blank?

Discussion

7. As above mentioned in the comment regarding the title, it is not clearly stated in the text whether the main focus of this study is on the challenges and unmet needs of the “individuals themselves” or those of the “caregivers”. This distinction could affect the content and interpretation in the Discussion.

8. The lack of a clear rationale for using two methods in this study makes the discussion ambiguous. It is unclear whether the main purpose of this study is (1) to understand broader issues using two methods, (2) to examine the consistency between the two methods, or (3) to demonstrate the significance of the NLP approach.

In the Introduction, the authors state, “Two studies applied ChatGPT-based TA, indicating that the manual TA and NLP-based thematic extraction were largely consistent (10) (11).” If the purpose corresponds to either (2) or (3) above, it is necessary to discuss the differences between the two methods in the current results.

Reviewer #3: Overall Evaluation

This study explores medical and psychosocial needs of caregivers of individuals with 22q11.2 deletion syndrome (22q11DS) using both manual thematic analysis (TA) and natural language processing (NLP)-based thematic extraction with an open-source large language model (LLM). The topic is highly relevant, the dataset is ethically collected, and the attempt to apply offline NLP to sensitive qualitative data is novel and valuable.

However, across the Introduction, Methods, Results, Discussion, and Limitations, the paper lacks methodological transparency and conceptual coherence between manual and NLP analyses. The main concerns relate to (1) the unclear positioning of NLP within the study’s aims, (2) the absence of quantitative or procedural validation for AI-derived results, and (3) over-generalization of AI outputs without critical reflection on model behavior.

Below, comments are organized by section, with major and minor issues clearly indicated.

Major Points

1. Clarity of Study Aim and Role of NLP

(Section: Introduction, last two sentences)

The introduction does not clearly articulate whether the study aims to validate NLP as a replacement for manual TA, or to demonstrate feasibility of automatic extraction.

Clarify if NLP serves as (a) a methodological validation exercise, (b) a complement to human TA, or (c) an exploratory tool for scalability.

State explicitly how this objective shapes the study design and evaluation.

2. Reliability and Transparency of NLP Outputs

(Section: Methods - NLP-based thematic extraction; Discussion, paragraphs 4-5)

While the study emphasizes “fully automated extraction,” the process of theme naming, editing, and validation remains opaque.

Specify which labels were generated verbatim by the model and which were rephrased or merged by researchers.

Describe any human-in-the-loop verification or quality control procedure.

Include representative raw outputs or prompt-response examples in the Supplement.

3. Evaluation of Concordance Between Manual and NLP Analyses

(Section: Discussion, first paragraph)

The discussion claims that both approaches identified “common themes,” but does not quantify or operationalize concordance.

Provide a correspondence table between manual and NLP categories or calculate thematic overlap (e.g., percentage match, semantic similarity).

Distinguish whether differences arise from human interpretive bias or from NLP abstraction.

4. Critical Examination of NLP Abstraction and Model Behavior

(Section: Discussion, paragraph 1; Table 2 results)

NLP categories such as “Parental Support Through Social and Community Networks” generalize multiple caregiver statements into high-level concepts possibly absent from the original text.

Discuss whether such themes represent genuine conceptual synthesis or model hallucination.

Acknowledge that LLMs tend to over-generalize and produce normative language.

5. Granularity Gap Between Manual and NLP Themes

(Sections: Results – Manual TA vs. NLP results; Discussion, paragraph 2)

Manual TA yielded five fine-grained categories, while NLP produced three broad ones. The paper treats these as equivalent without clarifying analytical level.

Define the analytical layer used for comparison (micro vs. meso vs. macro themes).

If alignment occurs only at higher abstraction, explicitly say so.

6. Ambiguity of the Study’s Core Contribution

(Section: Discussion)

It is unclear whether the main contribution is (a) developing a privacy-preserving offline NLP workflow, (b) confirming LLM–TA consistency, or (c) discovering new 22q11DS-specific needs.

Reframe the concluding paragraph to emphasize the primary novelty—e.g., secure LLM use in qualitative healthcare research—while treating thematic findings as an illustrative case.

7. Incomplete Discussion of NLP Methodological Limitations

(Section: Limitations)

The Limitations section omits discussion of NLP-specific weaknesses: cultural and linguistic bias, prompt dependency, loss of contextual nuance, and difficulty attributing quotes to individuals.

Add explicit acknowledgment that LLMs may misinterpret idiomatic Japanese caregiver expressions or generate synthetic phrasing.

State that AI-based extraction is suitable for overview analysis but not for capturing emotional or contextual subtleties.

Minor Points

1. Reference Contextualization

(Introduction & Discussion)

When citing prior ChatGPT-based TA studies (refs 10, 11, 22), specify how your approach differs - e.g., local processing, open-source model, or ethics compliance. A concise comparison table would improve clarity.

2. Positionality Statement Integration

(Discussion, paragraph 1 middle)

The authors mention “researchers’ positionality shaped by clinical experience,” but this appears abruptly. Move this to Limitations or Methods and explicitly state how positionality informed interpretation.

3. Participant Representation and Bias

(Limitations)

The sampling bias description is appropriate but could specify which demographic or clinical characteristics are overrepresented. If known, briefly compare responders and non-responders.

4. Translation and Language Transparency

(Methods – data processing; Results tables)

Since qualitative data were originally in Japanese, describe translation workflow (e.g., who translated, whether back-translation was used). This is vital for replicability and cross-cultural interpretation.

5. Terminological Consistency

(Throughout Results & Discussion)

Unify expressions such as “comprehensive support,” “comprehensive improvement,” and “integrated care.” Consistent terminology will make the comparison between manual and NLP themes clearer.

Check for other inconsistent expressions throughout the manuscript as well.

6. Minor Editorial Corrections

Correct typographical error “22q12DS” -> “22q11DS” in Limitations.

Improve table formatting so Tables 1 and 2 share identical columns (“Theme / Subtheme / Illustrative Quote / Participant ID”).

If “22q12DS” actually exists as a distinct condition, briefly explain it to avoid confusion.

Concluding Assessment

The manuscript addresses an important intersection of qualitative health research and natural language processing. Its strengths lie in ethical rigor and methodological innovation through offline LLM deployment. Nonetheless, the argumentation would benefit from tighter alignment between aims, methods, and interpretations.

Specifically, the paper should:

1. Clarify the analytic goal of combining TA and NLP,

2. Quantitatively or systematically evaluate thematic concordance,

3. Disclose the extent of human intervention in AI outputs, and

4. Critically reflect on the abstraction tendencies and cultural limitations of LLMs.

Addressing these points will greatly enhance methodological transparency, theoretical coherence, and the overall scientific contribution of the study.

**Do you want your identity to be public for this peer review?** For information about this choice, including consent withdrawal, please see our Privacy Policy .

Reviewer #1: No

Reviewer #2: No

Reviewer #3: No

---

## [Decision Letter · Decision Letter 1]

4 Mar 2026

Medical challenges and unmet needs of individuals with 22q11.2 deletion syndrome as perceived by caregivers: A thematic analysis and natural language processing-based thematic extraction

PMEN-D-25-00414R1

Dear Prof Kasai,

We are pleased to inform you that your manuscript 'Medical challenges and unmet needs of individuals with 22q11.2 deletion syndrome as perceived by caregivers: A thematic analysis and natural language processing-based thematic extraction' has been provisionally accepted for publication in PLOS Mental Health.

Best regards,

Wenwang Rao

Academic Editor

PLOS Mental Health

Reviewer Comments (if any, and for reference):

Reviewer's Responses to Questions

**Comments to the Author**

Reviewer #1: All comments have been addressed

Reviewer #2: All comments have been addressed

Reviewer #3: All comments have been addressed

publication criteria ? Is the manuscript technically sound, and do the data support the conclusions? The manuscript must describe methodologically and ethically rigorous research with conclusions that are appropriately drawn based on the data presented.

Reviewer #1: Yes

Reviewer #2: Yes

Reviewer #3: Yes

3. Has the statistical analysis been performed appropriately and rigorously?

Reviewer #1: Yes

Reviewer #2: Yes

Reviewer #3: Yes

4. Have the authors made all data underlying the findings in their manuscript fully available (please refer to the Data Availability Statement at the start of the manuscript PDF file)?

Reviewer #1: Yes

Reviewer #2: No

Reviewer #3: Yes

5. Is the manuscript presented in an intelligible fashion and written in standard English?

Reviewer #1: Yes

Reviewer #2: Yes

Reviewer #3: Yes

Reviewer #1: Thank you for your careful and thoughtful revision. The manuscript has improved substantially, and I appreciate the clear efforts made to address the previous comments. In particular, the revised version now provides a much stronger methodological rationale, improved transparency regarding both the manual and NLP-assisted analyses, a clearer and more balanced comparison between approaches, and a more appropriately cautious discussion of the study’s contribution and limitations.

Reviewer #2: I checked the authors' response and the revised manuscript, and confirmed that the authors have appropriately addressed my comments.

I think the revised paper has substantially improved in clarity and in the quality.

Reviewer #3: I have reviewed all the corrections and confirmed that they have been implemented correctly. Therefore, I hereby accept them.

**Do you want your identity to be public for this peer review?** For information about this choice, including consent withdrawal, please see our Privacy Policy .

Reviewer #1: No

Reviewer #2: No

Reviewer #3: No
